# Multimodality Imaging of Benign Primary Cardiac Tumor

**DOI:** 10.3390/diagnostics12102543

**Published:** 2022-10-20

**Authors:** Yixia Lin, Wenqian Wu, Lang Gao, Mengmeng Ji, Mingxing Xie, Yuman Li

**Affiliations:** 1Department of Ultrasound Medicine, Union Hospital, Tongji Medical College, Huazhong University of Science and Technology, Wuhan 430022, China; 2Clinical Research Center for Medical Imaging in Hubei Province, Wuhan 430022, China; 3Hubei Province Key Laboratory of Molecular Imaging, Wuhan 430022, China

**Keywords:** multimodality imaging, benign primary cardiac tumor

## Abstract

Primary cardiac tumors (PCTs) are rare, with benign PCTs being relatively common in approximately 75% of all PCTs. Benign PCTs are usually asymptomatic, and they are found incidentally by imaging. Even if patients present with symptoms, they are usually nonspecific. Before the application of imaging modalities to the heart, our understanding of these tumors is limited to case reports and autopsy studies. The advent and improvement of various imaging technologies have enabled the non-invasive evaluation of benign PCTs. Although echocardiography is the most commonly used imaging examination, it is not the best method to describe the histological characteristics of tumors. At present, cardiac magnetic resonance (CMR) and cardiac computed tomography (CCT) are often used to assess benign PCTs providing detailed information on anatomical and tissue features. In fact, each imaging modality has its own advantages and disadvantages, multimodality imaging uses two or more imaging types to provide valuable complementary information. With the widespread use of multimodality imaging, these techniques play an indispensable role in the management of patients with benign PCTs by providing useful diagnostic and prognostic information to guide treatment. This article reviews the multimodality imaging characterizations of common benign PCTs.

## 1. Introduction

Cardiac tumors are neoplasms that can involve all of heart structures including the myocardium, valves, and cardiac chambers. The cardiac tumor was first described by the pathologist Realdus Columbus, but it was not diagnosed for the first time in a living patient until 1934 [1,2]. In 2015, the World Health Organization (WHO) presented a new classification of primary cardiac tumors (PCTs) separating into benign tumors, uncertain biological behavior tumors, germ cell tumors, and malignant tumors [3]. PCTs are exceedingly rare entities. Based on the autopsy series, the incidence of PCTs was 0.002–0.3% and the prevalence of PCTs was 0.001–0.03% [4,5,6]. Benign PCTs are more common than malignant ones at any stage in life. In adults, approximately 75% of cardiac tumors are benign and nearly half of these tumors are myxomas [7]. Among children, up to 90% of primary neoplastic tumors are benign. Rhabdomyoma and fibroma are the most common subtypes in the pediatric population, accounting for about 80% of all benign PCTs [8]. Before the application of noninvasive imaging modalities to the heart, our understanding of benign PCTs is largely limited to case reports and autopsy studies. Clinicians have been dealing with increasing numbers of benign PCTs in living patients over the past few decades [9], which primarily attributes to the advances in cardiac imaging techniques [10,11,12]. While the histology is benign and the prognosis is better than those of malignancy, benign PCTs may lead to life-threatening events by affecting hemodynamics and causing malignant arrhythmia and embolism [13,14,15,16]. With the widespread use of multimodality imaging, these techniques provide useful diagnostic and prognostic information to guide treatment prior to adverse outcomes and open biopsy, making their role invaluable in the management of patients with benign PCTs and improving patient quality of life.

## 2. Clinical Manifestations

Patients with benign PCTs are often asymptomatic or these masses are discovered incidentally during evaluation for seemingly unrelated nonspecific symptoms. These symptoms are mainly dependent on their location, size, mobility, friability, and relation to adjacent cardiac structures and can be categorized as systemic or constitutional manifestations, systemic embolization, and cardiac symptoms resulting from intracardiac obstruction of tumors [17,18,19].

Systemic manifestations: benign PCTs can cause constitutional symptoms including fever, fatigue, weight loss, coughing, muscle pain, or arthralgia, which may be confused with infective endocarditis [20].

Cardiac symptoms: direct tumor effects can be further divided into obstruction and arrhythmia [21]. Intracardiac tumors can restrict blood flow to the heart, interfering with valvular function, which is associated with symptoms of obstruction, leading to syncope and sudden death. When neoplasms are located in the right heart, they can lead to symptoms of right heart failure and mimicking tricuspid stenoses, such as ascites, lower extremity edema, and superior vena cava syndrome [22,23]. In the event of tumors in the left heart, patients develop symptoms related to pulmonary edema and mitral valvular obstruction, including dyspnea, shortness of breath, chest discomfort, and syncope [17,24]. Cardiac tumors can interfere with the conduction system or disrupt normal myocardium, leading to arrhythmia. Any arrhythmia can occur, including atrioventricular block, ventricular tachycardia, pre-excitation, atrial fibrillation and ventricular fibrillation, and even cardiac arrest in severe cases [16,25,26,27].

Embolisms: according to the location of embolism, it can be divided to pulmonary and systemic embolism phenomenon. Tumors involving the left heart may result in systemic embolization, leading to stroke, retinal artery emboli, myocardial infarct, splenic or renal infraction, mesenteric ischemic or acute limb ischemic [17,21,28,29]. The cerebrovascular system is the most common site to be affected, presenting as ischemic stroke [30]. Pulmonary embolization is typically caused by a mass in the right heart [31]. In rare cases, a right-side tumor can cause systemic embolization through right-to-left cardiac shunting. 

## 3. Multimodality Imaging

Benign PCTs have a favorable prognosis and can be curable by complete excision but may lead to significant complications and even sudden death if there is no accurate diagnosis and treatment in a timely fashion [32]. The evaluation of these masses is not simple, involving varieties of diagnostic techniques such as non-invasive imaging techniques and tissue histology. Although invasive histological biopsy is the gold standard for diagnosis, there are risks of potential complications such as pneumothorax, embolism, pericardial tamponade, valve damage, vascular injury, and induced arrhythmias [10,33]. The emergence and development of modern imaging technologies have enabled the non-invasive diagnosis of cardiac tumors. Echocardiography, cardiac magnetic resonance (CMR), cardiac computed tomography (CCT), and positron emission tomography (PET) play a significant role in the accurate diagnosis and characterization of lesions. The initial aims of multimodality imaging assessment are to determine the presence and the precise location of the mass in the heart, which is crucial for further evaluation and treatment planning [34]. Multimodality imaging contributes to early detection, early diagnosis, and timely effective treatment of cardiac benign tumors, and has improved the approach to diagnosis and management of tumors [35,36]. In essence, because each imaging method has its own advantages and disadvantages, multimodality imaging uses two or more imaging types to provide valuable complementary information.

### 3.1. Echocardiography

Transthoracic echocardiography (TTE) is often the preferred imaging modality for the evaluation of cardiac PCTs because of its low cost and wide availability. Additional advantages include non-invasive, lack of radiation exposure, fast, safe, and describing heart structure in a variety of imaging planes [37,38]. In the case of getting good images, TTE allows for accurate assessment of tumor characteristics such as size, shape, attachment, mobility, location as well as hemodynamic consequences (e.g., valvular obstruction secondary to the tumor, mimicking tricuspid or mitral stenosis). What is more, it is the optimal imaging technique for the evaluation of small mobile masses (<1 cm) and valvular masses because of its high temporal and spatial resolution [39]. However, due to the limitation of imaging planes, TTE is unable to provide a comprehensive assessment of right heart, mediastinal and extracardiac structures outside of standard imaging views. Other limitations of TTE include the potential poor acoustic windows, especially for patients with obesity or chronic lung disease, as well as the lack of the capacity for tissue characterization [40]. If the image quality of TTE is poor, transesophageal echocardiography (TEE) may be considered to further evaluate tumors. The transducer of TEE, with high frequency, is located in the esophagus and closes to the posterior of the heart, so that sound waves are not necessary to penetrate the chest wall. Therefore, TEE can obtain optimal imaging quality for cardiac tumors. Additionally, the far superior resolution of TEE compared with TTE resulted in some of the detailed information of the tumor (such as calcification) being easier to see with TEE [41]. What is more, TEE can better characterize the size, appearance, morphology, location, and attachment point of the benign tumor, especially in the left atrium (LA), LA appendage, right atrial (RA), and cardiac valve [42,43]. In particular, tumors attached to the mitral and aortic valves can be sensitively detected, because the esophagus is directly posterior to the LA. In addition, TEE also provides a better characterization of valvular stenosis and leaks, which is meaningful for deciding the necessity and timing of surgery. The development of three-dimensional echocardiography (3DE) provides an incremental imaging modality for a more accurate evaluation of cardiac tumors. Three-dimensional echocardiography, based on a large number of sectional views to reconstruct the mass and cardiac structures, resolves the limitation of geometric assumptions and enables a more accurate assessment of volume, shape, attachment site, and the relationship of the tumor to adjacent structures [44]. Contrast echocardiography is increasingly used to assess cardiac tumors and can improve detection sensitivity in patients with poor acoustic windows. It is an emerging approach for assessing myocardial perfusion and the relative perfusion of a cardiac tumor. Furthermore, contrast echocardiography can evaluate the vascularity of tumors according to the difference in perfusion. For example, compared with adjacent myocardium, perfusion of benign PCTs is relatively lower because of poor blood supply, and the opposite is true for malignancy [45,46,47].

### 3.2. Cardiac Magnetic Resonance

Although echocardiography is the first-line imaging modality used to assess a suspected cardiac tumor, when information is insufficient, CMR can further evaluate the mass and provide incremental information. Because of its adequate spatial resolution, the ability of multiplanar image reconstruction, excellent soft tissue characterization, and the capacity to discriminate different tissue characteristics, CMR has become an indispensable method for the comprehensive evaluation of cardiac tumors and guidance for clinical diagnosis and treatment [48,49]. CMR has a variety of imaging sequences, including T1-weighted imaging (T1WI), T2-weighted imaging (T2WI), resting first-pass perfusion, steady-state free procession sequencing, early gadolinium enhancement (EGE), late gadolinium enhancement (LGE) [50]. T1WI can descript the tissue characteristics of cardiac tumors. T2WI can not only describe the tissue characteristics of tumors but also determine the necrosis within the tumor. SSFP sequencing can acquire images quickly and has a high signal-to-noise ratio, so it can better distinguish between blood pools and the endocardium. EGE is rarely used to characterize tumors and is primarily used to identify thrombus. LGE can be used to assess the enhancement features of tumors and to analyze myocardial infiltration. The vascularity of the tumor and heterogeneous areas within the tumor can be assessed by first-pass perfusion [51]. In conclusion, CMR can be used to identify characteristics relevant to histopathology, such as calcification, fat infiltration, necrosis, fibrosis, fluid, hemorrhage, cystic changes, and ferrous infiltration within tumors [52,53]. In addition, CMR allows for the determination of location, homogeneity, morphology, extension, border, mobility of cardiac masses, and valvular dysfunction. What is more, CMR can evaluate the relationship between tumor and lung, pericardium, or mediastinum. Contrast enhancement can describe the vascularity of the tumor and its relationship to blood vessels. The tissue characterization of tumors, cardiac function and hemodynamics, and the relationship between cardiac tumors and extracardiac structures provided by CMR are of great importance in tumor treatment and prognosis assessment [48,54,55,56]. However, because of its lower temporal resolution compared to an echocardiographic approach, CMR is inapplicable to evaluating small highly mobile cardiac or valvular tumors, which are almost invisible if they are less than 10 mm in diameter [55,57]. An additional disadvantage of CMR is the need for electrocardiographic gating, which prevents analyzable images from being obtained in patients with significant arrhythmias [58]. Other limitations include high price, long acquisition times, inability to be used in patients who cannot hold their breath or are hemodynamic instability, contraindications that include claustrophobia, and patients with implanted cardiac devices such as pacemakers or intracardiac defibrillators [59,60].

### 3.3. Cardiac Computed Tomography

With the increased use of CCT for the evaluation of cardiac masses, it has become a second-line diagnostic modality for the assessment of cardiac masses, especially when other imaging modalities are contraindicated or inadequately evaluated [61]. With the emergence and advance of technologies such as multidetector CT, helical CT, and electrocardiographic (ECG)-gated CT, CCT has achieved submillimeter spatial resolution, shorted scanning time, minimized motion-related artifacts and improved temporal resolution and imaging quality, which is helpful for more accurate description of cardiac masses [62]. CCT has the ability to characterize tissue by the evaluation of density and perfusion. The use of contrast-enhanced CT is particularly helpful in establishing the differential diagnosis of cardiac masses, assessing the vascular distribution and fibrous component of tumors, or demonstrating vascular malformation [63]. Compared with other imaging modalities, CCT is the preferred and optimal imaging technique for the evaluation of cardiac tumor calcification and other non-cardiac structures in the chest [64,65]. Because of its excellent spatial resolution unmatched by other modalities, CCT can precisely and comprehensively evaluate the relationship between cardiac tumors and the myocardium, pericardium, chest, lung, cardiac valves, and corresponding vascular structures. In the presence of suspected coronary artery disease, coronary angiography should be performed to exclude the tumor adjacent to or causing obstruction to the coronary arteries, which is sometimes helpful to make a rational surgical plan [66]. Limitations of CCT include the presence of radiation exposure, the risk of contrast-induced adverse events, lower temporal resolution compared with echocardiography or CMR, and lower soft-tissue resolution compared with CMR. With the development of technology, prospective ECG-gating has emerged to minimize radiation exposure [67,68]. In clinical practice, echocardiography and CMR are the first-line primary imaging modalities for the diagnosis and management of cardiac tumors, whereas CCT is a powerful and valuable complementary tool.

### 3.4. Positron Emission Tomography

Although anatomical imaging modalities, such as echocardiography, CCT, and CMR, can accurately assess the location, morphology, and margin of tumors, their ability to provide metabolic information is limited. PET can assess the metabolic activity of tumors by using ^18^F-fluorodeoxyglucose (^18^F-FDG) to visualize cell metabolism [69]. Fusion imaging of CT with ^18^F-FDG PET (^18^F-FDG PET/CT) can provide both anatomical and metabolic information, which is helpful to the diagnosis, stage, treatment, and prognosis of cardiac tumors and strengthen the detection of occult distant metastases [34,70,71,72]. Quantification of FDG uptake, based on maximum standardized uptake value (SUVmax), can help to distinguish benign from malignant cardiac tumors. In general, benign PCTs have no FDG uptake or only a slight uptake, whereas this parameter is relatively high in malignant tumors [34,73]. Nensa et al. demonstrated that the SUVmax of benign tumors is usually less than 5.2, while most malignancies are above this cutoff, with a sensitivity of 100% and specificity of 92% [74]. However, it is worth noting that cases of benign PCTs with hypermetabolism have been reported [75,76]. ECG-gated PET was selected as far as possible to collect cardiac tumor images, in order to minimize motion artifacts and improve the sensitivity of detecting lesions, especially small or low uptake lesions [77,78]. The limitations include that cardiac lesions require a longer fasting period and a more restrictive diet compared to other lesions. Another disadvantage of PET is the presence of radiation exposure [79,80,81].

## 4. Benign Cardiac Primary Cardiac Tumor (Table 1)

### 4.1. Myxoma

Myxoma (Figure 1) is the most frequent benign PCT in adults, making up 50–70% of them, but is relatively rare in children, representing only 10% of benign PCTs in the pediatric population [6,9,24,82,83]. It is an intracavitary tumor derived from multipotent mesenchymal cells in the subendocardial tissue, which mainly occurs in middle age with 70% predominance for females [17,84,85]. Cardiac myxomas can arise from any heart cavities, but LA is the most frequently involved site, 75% in LA, 20% in RA, and around 3–4% in the left ventricle (LV) and right ventricle (RV), respectively [1,83,86]. Approximately 90% of myxomas are attached to the fossa ovalis by a stalk and other less common anatomical origins include inferior vena cava (IVC), the atrial free wall, and the valve leaflets [32,64]. Up to 90% of myxomas are solitary and sporadic, and less than 10% are multiple and familial. The latter is associated with the Carney complex, an autosomal dominant multiple neoplasia and lentiginosis syndrome, and usually affects younger patients [24,87]. It is seen in the appearances, cardiac myxomas can be divided into polypoid lesions (smooth surface and hard) and papillary lesions (soft, gelatinous, and fragile) [17]. Large polypoid myxoma may cause obstructive symptoms, and conversely, papillary lesion tends to present as embolism. In rare cases, they undergo calcification or ossification. Clinical symptoms mainly depend on the location, size, and friability of tumors, and the typical triad includes embolism, obstruction, and constitutional symptoms [88,89]. Surgical resection is usually required when patients present with clinical symptoms.

**Table 1 diagnostics-12-02543-t001:** Features of benign primary cardiac tumor.

Type of Tumor	Common Age	Common Gender	Most Common Site	Echo Features	CT Features	CMR Features
Myxoma	Middle age (30–60 y); younger if associated with Carney complex	Females	Left atrium; attached to the fossa ovalis	Heterogeneous; narrow pedicle; regular mobile throughout the cardiac cycle	Heterogeneous; low-attenuation; calcification seen in 10–20% of patients	Smooth, well-defined, lobular or oval; heterogeneous; isointense on T1WI; hyperintense on T2WI; heterogeneous enhancement
Papillary fibroelastoma	40–80 y	Men	Cardiac valves	Small (usually <1.5 cm); round; well-circumscribed; homogeneously textured appearance; a short pedicle; shimmering edges	Difficult to observe	Difficult to observe
Lipoma	No defined age distribution	No defined sex distribution	Left heart; subendocardial layer	Usually hypoechoic in the pericardial space, homogenous and hyperechoic in cardiac chambers; broad base; immobile; well-circumscribed	Homogeneous fat attenuation; well-defined; smooth; encapsulated; no contrast enhanced	Homogeneous hyperintense on T1WI and complete signal loss in fat suppression sequence; hyperintensity on T2WI; no enhancement
Rhabdomyoma	Infants and children	No defined sex distribution	Usually intramyocardial or intracavitary; no difference in distribution between the left and right heart	Multiple; small; round; lobulated; well-circumscribed; homogenous hyperecho	Multiple homogeneous low attenuation; no enhancement	Isointense on T1WI; hyperintense on T2WI; no enhancement
Fibroma	Infants and children	No defined sex distribution	Ventricles	Large; intramural; well-circumscribed; noncontractile; central calcification	Homogenous; intramural; soft-tissue attenuation; central calcification; little to no enhancement	Iso-intense on T1WI; hypointense on T2WI; homogenous; no enhancement on resting first-pass perfusion imaging and EGE; hyperenhancement on LGE
Paraganglioma	20–60 y (typically in young adults)	No defined sex distribution	Left atrium; interatrial septum; aortic body	Granular; oval; well-demarcated; broad base; heterogeneous	Well-circumscribed; heterogeneous; low attenuation; heterogeneous marked enhancement	Isointense or hypointense on T1WI; hyerointense on T2WI; heterogeneous and peripheral rim enhancement
Hemangioma	Adulthood	Females	Ventricles	Well-circumscribed; oscillated with the cardiac cycle; blood flow signals on color Doppler flow imaging; obviously enhancement	Well-defined; low density or equal density; heterogeneous intense enhancement; “vascular blush” on coronary angiography	Heterogeneous isointense or hypointense on T1WI; hyperintense on T2WI; heterogeneous enhancement

Echocardiography is the primary imaging method for the diagnosis and management of cardiac myxomas. CCT and CMR play complementary roles, providing incremental value. On echocardiography, cardiac myxoma typically presents as a mobile heterogeneous echogenic mass attached to the endocardial surface (usually the fossa ovalis) by a narrow pedicle [90]. Heterogeneity is a common characteristic of cardiac myxomas because they may contain varying amounts of myxoid, hemorrhagic, necrotic, ossific, and cystic tissue. Myxomas, usually round or oval in shape, are mainly located in the LA. LA myxomas move regularly throughout the cardiac cycle, which protrudes into LV through the mitral valve orifice in diastole, leading to stenosis of the mitral valve orifice, and return to the LA in systole [91]. What is more, echocardiography can provide accurate and intuitive information about cardiac myxomas, detailing their location, size, overall morphology, and motility. Three-dimensional echocardiography offers additional value in the assessment of myxomas by more accurately describing the spatial relationship between mobile myxomas and adjacent structures [92]. Three-dimensional echocardiography helps to analyze the specific attachment position of myxoma pedicle and mass heterogeneity by using the cropping function and digital analysis to anatomize the lesion [44]. Three-dimensional echocardiography can provide a better morphological characterization based on the microscopic appearance of the surface of myxoma, which plays an important role in its diagnosis and classification and has a good correlation with surgical and histopathological findings [93]. TEE is highly sensitive to the diagnosis of cardiac myxomas and more accurate to obtain the attachment point of these masses. TEE allows for better observation of the myxoma implantation site and identifies any potential masses invading the pulmonary or vena cava. Contrast echocardiography can be used to distinguish myxoma from thrombus. Thrombi are nonvascular masses, but myxomas have sparse blood vessels, with poor blood supply. After administration of ultrasound contrast agents, a complete lack of enhancement suggests thrombus, but myxomas generally tend to be partially or incomplete enhancement [47,94]. On CCT, myxomas are often detected as a heterogeneous low-attenuation intracavitary mass with lobular contour. Calcification is seen in 10–20% of patients and appears to be more common in RA than LA. Nevertheless, massive calcification is rare [64,95]. Contrast-enhanced CT demonstrates a distinct intracardiac well-defined spherical or ovoid mass with weak or absent enhancement. Cardiac myxomas appear as filling defects surrounded by enhancing intracardiac blood and are hypoattenuating or isoattenuating relative to the myocardium [96,97]. On CMR, myxomas usually manifest as a smooth, well-defined, lobular, or oval mass. They appear as heterogeneous appearance or isointense on T1WI and heterogeneous appearance or hyperintense on T2WI because of the composition of myxomas and their high extracellular water content [98,99]. The cine MRI images show the mobility of myxomas, which protrude into ventricles through the atrioventricular valves in diastole, resulting in obstruction to blood flow [100]. SSFP imaging can accurately assess the attachment point and location of myxomas. They appear as relatively hyperintense compared with the myocardium, and hypointense relative to the blood pool [100]. On resting first-pass perfusion images, myxomas may present as some slight heterogeneous enhancement, while 10–15 min after gadolinium contrast administration, cardiac myxomas may show patchy and more heterogeneous enhancement on LGE images [100]. On PET imaging, they may manifest as a mildly hypermetabolic hypodense area in heart cavities [101].

### 4.2. Papillary Fibroelastoma

Papillary fibroelastomas (PFEs) (Figure 2) are small (2–7 mm), with multiple papillary fronds, sea anemone-like appearance, and slow-growing intracardiac tumors attached to the endocardium by a short pedicle. PEFs have been considered the second most common benign PCTs and the most common tumors of the cardiac valves, accounting for 10% of all cardiac tumors and 75% of all cardiac valvular tumors [102,103]. With the improvement of imaging modalities and a better understanding of PEFs, cardiac PEFs are now thought to be probably the most common benign PCTs surpassing myxomas [104,105]. These tumors mainly affect men between the age of 40–80 years, and the average age of detection is about 60 years. PEFs can originate anywhere in the heart, but they usually occur in the cardiac valves, with more than 95% located in the left heart. The most commonly affected heart valve is the aortic valve (44%), followed by the mitral valve (35%), less frequently tricuspid and pulmonic valve (15% and 8%, respectively) [103,105]. They are more often detected downstream side of the valve, and usually do not lead to valvular dysfunction despite being attached to the valves. The LV endocardium is the most common non-valvular location [103]. Most of patients are asymptomatic, but some of patients may develop embolic events leading to systemic, coronary, or cerebral circulation obstruction [103]. Clinical manifestation is often insidious or non-specific, which leads to delayed diagnosis and treatment [106]. The current management of symptomatic patients is surgical resection, but the optimal management remains controversial, especially in asymptomatic patients [107].

On echocardiography, PFEs appear as small (usually <1.5 cm), round, with independent motion, echo-dense, and pedunculated valvular surface or endocardial masses with a short stalk attachment. Generally, they are well-circumscribed and homogeneously textured in appearance, having sometimes a speckled interior with stippling near the edges because of multiple papillary projections on their surface [108,109]. TEE is more sensitive in identifying small PFEs (<5 mm) compared with TTE. For patients with a negative TTE but high suspicion of the cardiogenic embolic phenomenon, TEE should be considered, which helps to strengthen the role of TEE in the evaluation of embolic events [109]. Three-dimensional echocardiography allows us to clearly visualize the location and the stalk attached to the valve leaflets or endocardial. Moreover, the anemone-like appearance with multiple papillary fronds could be better visualized on 3DE, which appears slightly stippled or shimmering on traditional echocardiography [110]. Because PEFs are small and attached to moving valves, they may be difficult to observe on CCT or CMR images. Occasionally, on ECG-gated CT images, they show up as a focal low attenuation mass with irregular borders, arising from a valve surface [66]. CCT enables visualization of the exact anatomic location of the PFEs attachment site and simultaneous evaluation of the coronary arteries. On CMR images, PFEs usually present as a round, small, homogeneous mass attached to valvular leaflets. PFEs demonstrate isointense signal intensity relative to myocardium on T1-weighted images, and hypointense signal intensity on T2-weighted images due to their high fibrous content, but part of the PEFs may also demonstrate a hyperintense signal on T2-weighted images [111,112]. On cine MRI images, they show hypointense signal intensity. LGE reveals that PEFs usually do not show late enhancement [24,111].

### 4.3. Lipoma

Cardiac lipomas are well-encapsulated benign tumors composed of mature adipocytes. They constitute approximately 10% of benign PCTs, and there is no defined age or sex distribution [113]. In general, lipomas are slow growing, whereas part of them show aggressive growth and may infiltrate into the myocardium [114]. Although lipomas can occur anywhere in the heart and may arise from all three layers of cardiac tissue, they are most frequently found on the left side of the heart [113]. About 50% of them arise from the subendocardial layer, and the other half originates from the subepicardial or myocardial layers and grow into the pericardium [115]. It is generally believed that most lipomas are asymptomatic or incidentally discovered, but occasionally might cause various symptoms ranging from dyspnea, chest pain, and fatigue to syncope and arrhythmia, and sudden cardiac death. Usually, subendocardial lipomas are small with a broad base and do not cause obvious symptoms. In contrast, subepicardial lipomas may be larger, which leads to anginal pain from compression of coronary arteries [24]. Smaller asymptomatic lipomas should be observed clinically, and generally do not require surgical resection unless severe symptoms [113].

The echocardiographic findings of lipomas vary according to their location. Pericardial lipomas may appear as a completely hypoechoic, partially hypoechoic, or completely echogenic mass, whereas intracavitary lipomas are homogenous and hyperechoic. The echocardiographic features of a cardiac lipoma are its broad base of attachment, without a narrow pedicle, immobile, well-circumscribed, and homogenous without evidence of calcification [116]. When liquefaction or necrosis occurs within a lipoma, a large hypoechoic area may be present [117]. CCT and CMR imaging, with high specificity, can accurately diagnose cardiac lipomas. On CCT images, they appear as a well-defined, smooth, encapsulated, homogeneous fat attenuation and no contrast-enhanced mass with or without linear septa [115]. Lipomas show the same signal intensity as subcutaneous fat in all CMR sequences. They present as homogeneous hyperintense compared with myocardium on T1WI and the characteristic complete signal loss of the mass is seen in the fat suppression sequence [118]. On T2WI, they tend to show hyperintensity relative to the myocardium. With gadolinium administration, lipomas do not demonstrate any enhancement. It is worth noting that because of the chemical shift effect, the black boundary sign in the cine sequence is helpful for the diagnosis of small lipomas. What is more, CMR can sensitively detect the myocardial infiltration of lipoma, which provides imaging evidence for treatment planning [118].

### 4.4. Rhabdomyoma

Rhabdomyoma (Figure 3), a congenital hamartoma, is the most prevalent benign PCT in infants and children, accounting for more than 60% of all PCTs and 50% of these patients associated with tuberous sclerosis [115]. Rarely is seen in adults. It can be diagnosed from before birth to 6 years of age, and the mean age is 2 weeks at the time of diagnosis [3]. Rhabdomyomas can be multiple in 90% of patients, usually involving the atria and ventricle with no difference in distribution between the left and right heart, and intraluminal extensions are present in up to 50% of patients [115]. Obstruction of the inflow or outflow tract caused by the mass protruding into the lumen may lead to symptoms of congestive heart failure. Arrhythmias are not rare, atrial or ventricular arrhythmias often occur, and may result in palpitations and syncopal symptoms. Since the natural course of most rhabdomyomas is spontaneous regression, the associated symptoms also undergo gradually disappear [3]. Therefore, conservative treatment can be carried out by serial echocardiography and ECG follow-up monitoring. Some reports have demonstrated that everolimus is a potential new therapeutic option for treating rhabdomyomas with significant clinical presentations [119]. Surgical intervention is usually reserved for patients with severe obstruction and intractable arrhythmias symptoms that are unresponsive to corresponding drugs [8].

On echocardiography, rhabdomyomas tend to be multiple small, round, lobulated, well-circumscribed solid masses in cardiac cavities. Sometimes myocardial embedding can be seen and presents as a homogenous hyperechoic mass of variable size, usually brighter than the surrounding myocardium [116]. Deformation imaging allows for distinguishing rhabdomyomas from fibromas. A rhabdomyoma, composed of myocytes with relatively elastic, deforms in the opposite direction to the surrounding myocardium. However, fibromas are constituted by noncompliant connective tissue, and they do not seem to deform or contract in any direction [120]. CCT often demonstrates multiple homogeneous low-attenuation intramural lesions with intracavitary extension. In contrast CCT, they generally show hypodense. On T1WI, rhabdomyomas tend to appear isointense to slightly hyperintense compared to the surrounding myocardium, and on T2WI reveal hyperintense. LGE typically shows there is no delayed gadolinium enhancement after the contrast material administration [99].

### 4.5. Fibroma

Composed of fibroblasts and connective tissue, cardiac fibroma (Figure 4) is considered as the second most common type of benign PCTs in children following rhabdomyoma, accounting for 12–16% of PCTs in pediatrics [121]. The average age at diagnosis is 13 years with nearly a third of patients being less than one year of age at presentation. However, only 15% of fibromas are detected in adults or the elderly [122]. They are noncapsulated solitary masses located in the myocardium, most commonly in the LV free wall (about 57%), followed by RV free wall (28%), interventricular septum (17%), and rarely involving the atria [121]. They are also associated with Gorlin (basal cell nevus) syndrome, which is more common in fibromas of atrial origin [123]. Cardiac fibroma often has no necrosis, hemorrhage, and cystic change, but central calcification is its feature reflecting poor blood supply of the tumor [124]. In general, the prognosis of benign PCTs is good, whereas cardiac fibromas have a relatively poor prognosis. Because fibromas involving the myocardium may interfere with conduction pathways and myocardial contraction, resulting in fatal arrhythmias, heart failure, and sudden death. These tumors do not show spontaneous regression, coupled with the risk of poor prognosis, so surgical treatment is recommended regardless of symptoms [16]. Fibroma can be treated by complete surgical resection, incomplete excision, or orthotopic heart transplantation based on its location, size, and resectability [125].

At echocardiography, cardiac fibromas typically present as a large (ranging from 3 to 10 cm in diameter), well-circumscribed, distinct, noncontractile, and solitary solid lesion within the myocardium [116,126]. Occasionally, central calcification can be observed. Contrast echocardiography reveals hypoperfusion relative to the surrounding myocardium. CCT images typically demonstrate a homogenous, intramural, and soft-tissue attenuation mass with occasionally central calcification with either sharply marginated or infiltrative. Central calcification is a relatively common feature of fibromas on CT, which contributes to their heterogeneous attenuation. Usually, these tumors present little to no contrast enhancement on contrast CCT. At CMR, fibromas manifest as iso-intense compared to the surrounding myocardium on T1WI and hypointense on T2WI due to their dense and fibrous nature. The signal intensity is typically uniform, but the presence of calcification may lead to central areas of low signal. Because of poor blood supply, fibromas do not enhance resting first-pass perfusion imaging. After administration of gadolinium contrast material, masses typically appear as no contrast enhancement on EGE imaging, whereas they demonstrate intense delayed hyperenhancement on LGE imaging due to their collagenous nature.

### 4.6. Cardiac Paraganglioma

Paragangliomas (PGLs) are rare neuroendocrine tumors, with rich vascularity with an incidence of 0.1–0.3%, arising from chromaffin cells located parasympathetic or sympathetic ganglia neural crest outside of the adrenal gland. Cardiac PGLs are even rarer, accounting for about 2% of all PGLs [124]. They affect women more frequently than men. The age of onset ranges from 20 to 60 years old but typically occurs in young adults [124]. Although cardiac PGLs can occur in all cardiac chambers, they are most commonly found in the LA in its posterior wall or roof, followed by the interatrial septum, and the aortic body, occasionally arising from the coronary artery. They can be divided into non-secretory (parasympathetic PGLs) or secretory (sympathetic PGLs) tumors. The former mainly produces compressive, embolization, or obstructive symptoms, while the latter secretes excessive catecholamines (norepinephrine, epinephrine, or dopamine) resulting in a plethora of corresponding symptoms, such as tachycardia, tremors, palpitations, flushing, hypertension, or hypotension) [127]. Most cardiac PGLs are benign, but up to 10% of them are malignant. Surgical resection is the definitive treatment and early complete surgical resection with clear and negative margins can lead to a full cure of tumors. If vessels and coronary arteries are extensively involved, surgical resection may be very difficult. Incomplete resection can lead to recurrence or metastasis, then continuous postoperative monitoring is very important.

Cardiac PGLs usually appear as a granular, oval, well-demarcated, and echogenic mass with a broad base of attachment at TTE. Sometimes, compression of adjacent structures can be seen such as the superior vena cava. On TEE, the encasement of coronary arteries can be assessed [124]. On CCT, cardiac PGLs are demonstrated as a well-circumscribed, heterogeneous lesion with low attenuation. Sometimes poorly defined margins can be found due to invasion or extracardiac extension. They are highly vascular neoplasms and are usually supplied by the left coronary artery (LCA), although occasionally dual blood supply from the left and right coronary arteries can be observed [128]. Contrast-enhanced CCT imaging reveals heterogeneous marked enhancement. Coronary angiography enables us to determine the relationship between cardiac PGLs and coronary arteries and the vascular supply to tumors. Typically, most of them show markedly increased signal intensity on T2WI, which helps to distinguish lesions from the surrounding cardiovascular structures [128]. T1WI shows isointense or hypointense compared to the myocardium, occasionally hyperintense due to hemorrhage within the tumor. On LGE imaging, the tumors are demonstrated to be intense heterogeneous and peripheral rim enhancement after injection of contrast material due to their vascularity and necrosis [124]. On PET imaging, cardiac PGLs appear as positive with intense uptake of radiotracers [129].

### 4.7. Hemangioma

Cardiac hemangiomas (Figure 5) are uncommon benign vascular tumors and constitute proximately 5–10% of all PCTs. They can occur in a wide age group, with a larger group present in adulthood and a smaller group affecting childhood [3]. Females are affected more frequently. The pathological feature of hemangiomas is the blood vessels with increasing vascularization are lined with benign proliferative endothelial cells. In addition, a large number of stromal elements such as myxoid, fat, and fibrous tissue can also be observed by pathological examination [115]. They can be divided into capillary, cavernous, or arteriovenous according to the principal type of proliferating vessels [130]. Typically, endocardial hemangiomas have histologic features of capillary or cavernous, and intramural hemangiomas resemble intramuscular hemangiomas in histologic analysis [3]. They are usually solitary and may affect any cardiac chamber. RA is the most common site in children, whereas ventricles are common in adults, especially in the lateral wall of the LV and less frequently in the anterior wall of the RV [3,115]. Patients are usually discovered incidentally with no symptoms until hemodynamics, coronary arteries, and conduction systems are affected or adjacent structures are compressed or invaded. Manifestations of the symptomatic patients are chest pain, arrhythmias, heart failure, dyspnea on exertion, syncope, stroke, pericardial effusion, cardiac tamponade, and even sudden death. At present, the effective treatment options for cardiac hemangiomas include medication therapies (corticosteroid, β-receptor blocker, and interferons) and surgical resection. Surgical resection is feasible in patients with clinical manifestations because of the potential risk of adverse outcomes, but it is controversial in asymptomatic patients [130,131]. Endocardial hemangiomas are well-defined, myxoid, and variably soft masses, whereas intramural hemangiomas are usually poorly circumscribed and cavernous masses with variably hemorrhagic or congested. Therefore, intramural hemangiomas are more difficult to excise than endocardial hemangiomas [3,131].

Cardiac hemangiomas manifest as a well-circumscribed echogenic mass that occasionally oscillates with the cardiac cycle at echocardiography. Color Doppler flow imaging reveals detectable blood flow signals in this tumor. Contrast echocardiography reveals hyper-perfusion relative to the surrounding myocardium. On unenhanced CCT, cardiac hemangiomas appear as a well-defined oval or round and low-density or equal-density mass. Sometimes, localized high-density shadows can be seen because of their fibers or phleboliths (calcified thrombi). On contrast-enhanced CT, they demonstrate as intense and heterogeneous. In general, after injecting of contrast agent, the central area of the tumor does not be immediately enhanced, but its delayed enhancement is of the same degree as the cardiac blood pool [131]. However, slow-flowing masses may show little to no enhancement. Currently, coronary angiography can be used to evaluate the presence of feeding vessels and the degree of vascularization of the tumor. The typical feature of hemangiomas is “vascular blush”, particularly in the capillary hemangiomas, which was detected in 80% of cases [132,133]. Cardiac hemangiomas generally present as heterogeneous intermediate signal intensity or hypointense signal intensity on T1WI and hyperintense on T2WI. Occasionally, areas of heterogeneous hypointense signal intensity can be described on T2WI. They manifest as heterogeneous enhancement or avid first-pass enhancement on resting first-pass perfusion images after injection of gadolinium contrast material [126,131].

## 5. Conclusions

Benign PCTs are rare in both children and adults, with most the common of them being rhabdomyomas in children and myxomas in adults. They are histologic benign masses and have a good prognosis but may lead to severe complications. Therefore, early diagnosis and treatment are necessary to improve the quality of life for patients. Multimodality imaging plays a crucial role in the evaluation of benign PCTs. Echocardiography remains the first-line method for describing these tumors, but CCT and CMR are complementary imaging modalities providing incremental information. With the advance in multimodality imaging techniques, we can more accurately characterize and diagnose benign PCTs as well as guide patient treatment. Therefore, a better understanding of the multimodality imaging characteristics of various benign PCTs helps to optimize the management of these tumors and further maximize the benefits for patients.

## Figures and Tables

**Figure 1 diagnostics-12-02543-f001:**
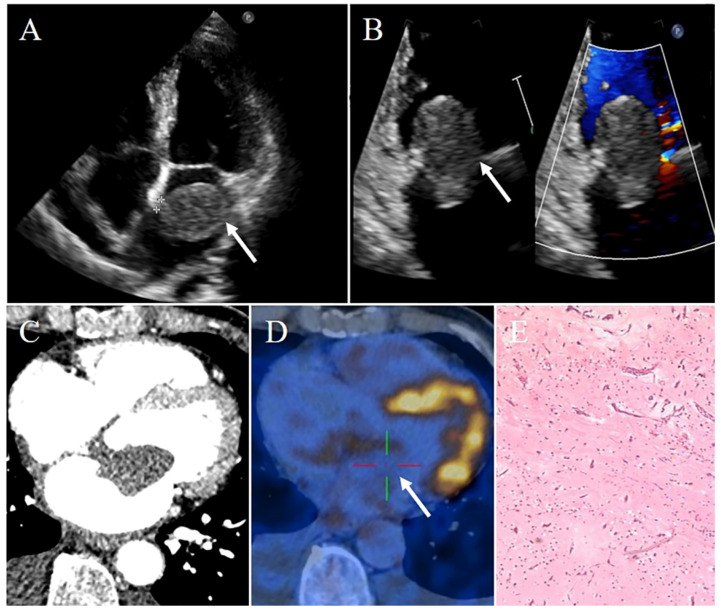
Left atrial myxoma in a 69-year-old man presenting with chest tightness and shortness of breath. (**A**) Transthoracic echocardiography showing a pedunculated mobile heterogeneous echogenic mass attached to the interatrial septum. This mass locates in the LA. (**B**) Part of this mass protruding into the LV through the mitral valve orifice in diastole, leading to stenosis of the mitral valve orifice. (**C**) Contrast-enhanced CT demonstrating a relative low density well-circumscribed mass originating from the interatrial septum, with absent of enhancement. (**D**) PET-CT imaging revealing the radionuclides slightly concentrated in the mass (SUVmax 4.6). (**E**) Pathology confirming myxoma. White arrows pointing to the left atrial myxoma and † marking stalk.

**Figure 2 diagnostics-12-02543-f002:**
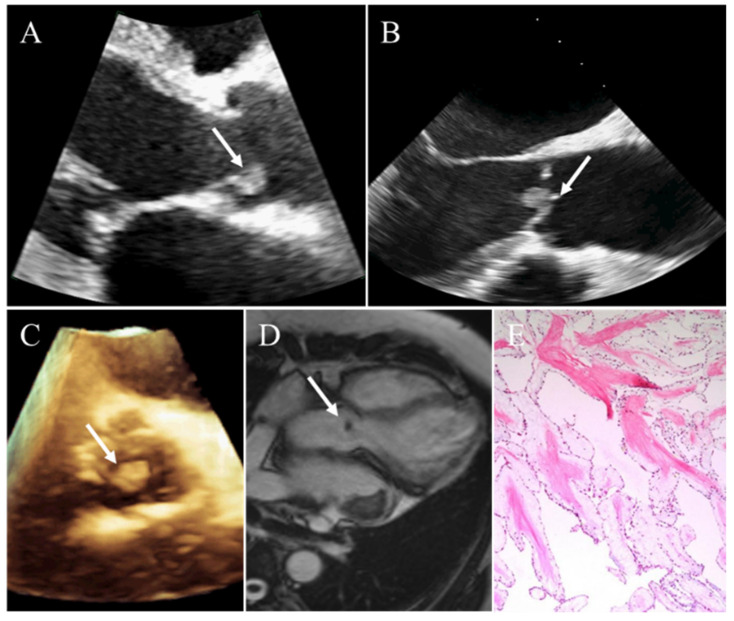
Incidental finding of a papillary fibroelastoma in a 62-year-old man. (**A**) A small mobile solid mass attaching to the aortic valve on transthoracic echocardiography. (**B**) Transesophageal echocardiography clearly showing the mass of aortic valve. (**C**) Three-dimensional echocardiography clearly visualizing the location and size of the mass. (**D**) CMR demonstrating a round, small, homogeneous mass attached to aortic valvular leaflet. (**E**) Pathology confirming papillary fibroelastoma. White arrows representing the papillary fibroelastoma.

**Figure 3 diagnostics-12-02543-f003:**
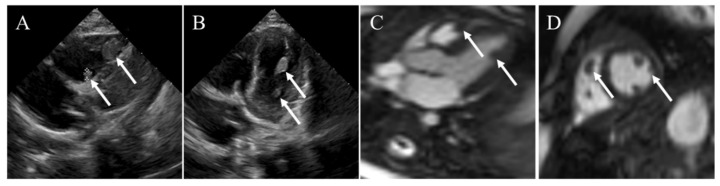
Rhabdomyoma incidentally found in a nine-month-old child. (**A**,**B**) Transthoracic echocardiography demonstrating multiple slightly hyperechoic masses in the LV and RV. (**C**,**D**) CMR confirming multiple biventricular masses. White arrows denoting the rhabdomyoma.

**Figure 4 diagnostics-12-02543-f004:**
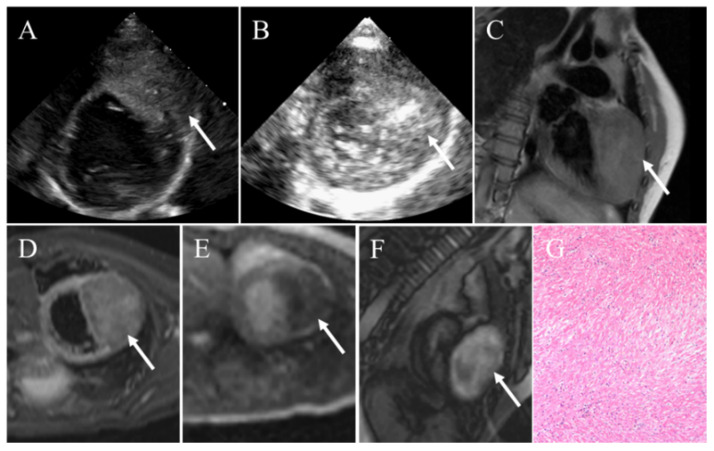
Cardiac fibroma in a 3-year-old child manifesting as palpitation and cough. (**A**) Transthoracic echocardiography demonstrating a large heterogeneous intramyocardial mass with sporadic calcific. (**B**) Contrast echocardiography revealing slight enhancement of contrast agent within the mass. (**C**) CMR showing an intramyocardial mass presenting iso-intense on T1-weighted images. (**D**) On T2-weighted images, the mass appearing slight hyper-intense. (**E**) The mass presenting as hypoperfusion on resting first-pass perfusion images. (**F**) LGE imaging revealing the mass appeared as obviously inhomogeneous high signal intensity relative to the myocardium. (**G**) Pathology confirming fibroma. White arrows pointing to the cardiac fibroma.

**Figure 5 diagnostics-12-02543-f005:**
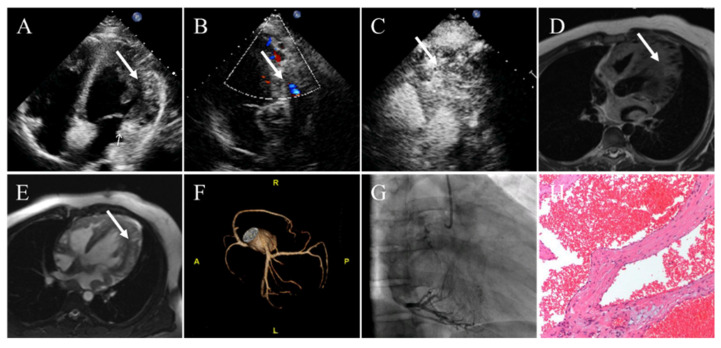
Cardiac cavernous hemangioma in a 16-year-old man presenting with palpitation. (**A**) Transthoracic echocardiography showing a heterogenous echogenic mass in the lateral wall of the LV. (**B**) Color Doppler flow imaging revealing coronary artery blood flow within the mass. (**C**) Contrast echocardiography demonstrating enhancement of contrast agent within the mass. (**D**) The left ventricular wall appearing inhomogeneous thickening with local nodules and diffuse edema on T2-weighted images. (**E**) Late gadolinium enhancement imaging demonstrating that the lateral wall of the left ventricle presented as obviously inhomogeneous hyperintense. (**F**) On contrast-enhanced CT, the coronary artery branches increased in the left ventricular myocardium. (**G**) Coronary angiography confirms the coronary arteries give off many branches and myocardial obviously staining in arterial phase. (**H**) Pathology confirming cavernous hemangioma. White arrows representing the cardiac cavernous hemangioma.

## Data Availability

Not applicable.

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
