# Peer review of "Multimodality Imaging of Benign Primary Cardiac Tumor"

_diagnostics, 2022, doi:10.3390/diagnostics12102543_

Round 1
Reviewer 1 Report
" Cardiac tumors are neoplasms that grow in the myocardium or adjacent tissues, which can involve papillary muscles or valves " the sentence is incorrect both from the Grammar point of view ("which" refers to "tissue"?) and from the cardiology point of view (what about tumors growing in the heart cavity or in the verses?)
"When neoplasms are located in the right heart, causing symptoms of right heart failure and tricuspid stenosis, such as ascites, lower extremity edema and superior vena cava syndrome". The verb of the principal clause is missing! nd the affirmation is not really true!
"Intramyocardial benign tumors may interfere with cardiac conduction system, leading to lethal arrhythmia, such as atrioventricular block, ventricular tachycardia, ventricular fibrillation and supraventricular- tachycardia, " This makes no sense! Interfering with the conduction system does not cause ventricular arrhythmias; supraventricular tachicardia is not a life-threatening arrhythmias!
What is a "preferable prognosis"?
"Although invasive histological biopsy is the gold standard for diagnosis, patients usually have experienced adverse events when tumors are diagnosed by this method." That's simply not true!
Figure 1: "PET-CT imaging revealing the radionuclides slightly concentrated in the mass (SUVmax 4.6).". And what about the myocardial uptake?
Author Response
Response to reviewer 1 comments:
Point 1: "Cardiac tumors are neoplasms that grow in the myocardium or adjacent tissues, which can involve papillary muscles or valves " the sentence is incorrect both from the Grammar point of view ("which" refers to "tissue"?) and from the cardiology point of view (what about tumors growing in the heart cavity or in the verses?)
Response 1: Thank you very much for your comments. We agree with your viewpoints. To avoid the misunderstanding, we have corrected the description of this sentence in the article. "Cardiac tumors are neoplasms that can involve all of the heart structures including myocardium, valves and cardiac chambers"(1).
Reference:
- Bussani R, Castrichini M, Restivo L, et al. Cardiac Tumors: Diagnosis, Prognosis, and Treatment. Curr Cardiol Rep. 2020;22(12):169. Published 2020 Oct 10. doi:10.1007/s11886-020-01420-z.
Point 2: "When neoplasms are located in the right heart, causing symptoms of right heart failure and tricuspid stenosis, such as ascites, lower extremity edema and superior vena cava syndrome". The verb of the principal clause is missing! nd the affirmation is not really true!
Response 2: Thank you very much for the comment. We are sorry for the grammatical error, and we have corrected the error in the revised manuscript. "When neoplasms are located in the right heart, they can lead to symptoms of right heart failure and mimicking tricuspid stenosis, such as ascites, lower extremity edema and superior vena cava syndrome".
Point 3: "Intramyocardial benign tumors may interfere with cardiac conduction system, leading to lethal arrhythmia, such as atrioventricular block, ventricular tachycardia, ventricular fibrillation and supraventricular- tachycardia, " This makes no sense! Interfering with the conduction system does not cause ventricular arrhythmias; supraventricular tachicardia is not a life-threatening arrhythmias!
Response 3: Thank you very much for your comment. We are very sorry for the misunderstanding caused by our improper expression and understanding. To avoid the misunderstanding, we have corrected the description of this sentence in the article. We agree with the point that supraventricular tachycardia is not a lethal arrhythmia. "Cardiac tumors can interfere with conduction system or disrupt normal myocardium, leading to arrhythmia. Any arrhythmia can occur, including atrioventricular block, ventricular tachycardia, pre-excitation, atrial fibrillation and ventricular fibrillation,"
Point 4: What is a "preferable prognosis"?
Response 4: Thank you very much for your comment. A "preferable prognosis" refers to "favorable prognosis". To avoid the misunderstanding, we have corrected the description of this phrase in the article.
Point 5: "Although invasive histological biopsy is the gold standard for diagnosis, patients usually have experienced adverse events when tumors are diagnosed by this method." That's simply not true!
Response 5: We thank the reviewer for the comment. Based on your comment, we have reviewed relevant literatures and revised this sentence in the article. "Although invasive histological biopsy is the gold standard for diagnosis, there are risks of potential complications such as pneumothorax, embolism, pericardial tamponade, valve damage, vascular injury, and induced arrhythmias". (1, 2)
Reference:
- Xie Y, Hong ZL, Zhao YC, et al. Percutaneous ultrasound-guided core needle biopsy for the diagnosis of cardiac tumors: Optimizing the treatment strategy for patients with intermural and pericardial cardiac tumors. Front Oncol. 2022;12:931081. Published 2022 Aug 5. doi:10.3389/fonc.2022.931081
- Oliveira GH, Al-Kindi SG, Hoimes C, et al. “Characteristics and Survival of Malignant Cardiac Tumors: A 40-Year Analysis of >500 Patients.” Circulation vol. 132,25 (2015): 2395-402. doi:10.1161/CIRCULATIONAHA.115.016418.
Point 6: Figure 1: "PET-CT imaging revealing the radionuclides slightly concentrated in the mass (SUVmax 4.6)". And what about the myocardial uptake?
Response 5: Thank you very much for your comment. The uptake of myocardium is normal (SUVmax 1.7).

Reviewer 2 Report
The revied paper is a comprehensive review discussing multimodality imaging to detect and treat primary cardiac tumors.
This review is suggested for a focused issue about noninvasive diagnosis of cardiac tumors. It is well-built and comprehensive. I think the authors' way of organizing this huge subject is sensible and makes this review easy to follow.
I have several concerns and comments:
1. there are many grammatical errors which make the paper harder to read and follow. Most of them are in the abstract and the first three topics. for example:
in the abstract - line 14 "Primary tumors are rare, of which...". Line 15: "Patients with benign PCTs are usually asymptomatic and are found incidentally..." (the subject is the tumor, not the patient). Line 18: "varies" should spell "various". In paragraph 1 - line 35 "According to the 2015 WHO presented a new classification...", the first two words are unnecessary. etc.
there is a need for English language revision.
2. In the description of the different imaging modalities, in the discussion about TEE, there is a need to better describe the added benefit it has over TTE. Fir which valves does it give a better picture, and why?, I also suggest mentioning the better characterization of valvular stenosis and leaks, which is meaningful for deciding the necessity and timing of surgery.
While discussing MRI, I suggest the authors add a short explanation about T1, T2, SSFP sequencing, EGE and LGE, and preferably another sentence about special sequences such as fat-sat and tagging which are important for tumor identification.
3. The paragraphs discussing each tumor are well written and easy to follow. The figures are informative and compliment the text. There is a brief description of major echo, CT and MRI findings. Cardiac PET is not discussed in all of the tumor types. I think that a table can be beneficial in summarizing the text (tumor type, common age and gender, common locations and appearance on echo, CT, MRI)
Although malignant primary cardiac tumors are extremely rare, and are not within the scope of this paper, I believe the authors should mention the common types, and review papers discussing the characteristics of malignant primary cardiac tumors.
Author Response
Response to reviewer 2 comments:
Point 1: There are diffuse, scattered typos that should be corrected.
Response 1: Thank you very much for your suggestion. I'm sorry for these errors in the article, and we have carefully corrected these errors in the revised manuscript.
Point 2: Introduction: Line 34: “Cardiac tumor was first described by the pathologist Realdus Columbus, but it was not diagnosed for the first time in a living patient until 1934.” Reference is missing
Response 2: Thanks for your careful review. We have carefully examined this sentence, and have added references 1 and 2 to the end of this sentence.
Point 3: Multimodality Imaging: “Although invasive histological biopsy is the gold standard for diagnosis, patients usually have experienced adverse events when tumors are diagnosed by this method.” Reference is missing
Response 3: Thanks for your careful review. We have carefully examined this sentence, and have added references 10 and 33 to the end of this sentence.
Point 4: Papillary Fibroelastoma:
“With the improvement of imaging modalities and better understanding of PEFs, cardiac PEFs are now thought to be probably the most common benign PCTs surpassing myxomas.” Reference is missing
“The most commonly affected heart valve is the aortic valve (44%), followed by the mitral valve (35%), less frequently tricuspid and pulmonic valve (15% and 8%, respectively).” Reference is missing
Response: Thanks for your careful review. We have carefully examined these two sentences, and have added references 104, 105 and 103, 105 to the end of those sentence respectively.

Reviewer 3 Report
Well written review about multimodality imaging of benign primary cardiac tumor
I have some comments:
1)There are diffuse, scattered typos that should be corrected.
2)Introduction: Line 34: “Cardiac tumor was first described by the pathologist Realdus Columbus, but it was not diagnosed for the first time in a living patient until 1934.” Reference is missing
3)Multimodality Imaging: “Although invasive histological biopsy is the gold standard for diagnosis, patients usually have experienced adverse events when tumors are diagnosed by this method.” Reference is missing
4)Papillary Fibroelastoma:
“With the improvement of imaging modalities and better understanding of PEFs, cardiac PEFs are now thought to be probably the most common benign PCTs surpassing myxomas.” Reference is missing
“The most commonly affected heart valve is the aortic valve (44%), followed by the mitral valve (35%), less frequently tricuspid and pulmonic valve (15% and 8%, respectively).” Reference is missing
Author Response
Response to reviewer 3 comments:
Point 1: there are many grammatical errors which make the paper harder to read and follow. Most of them are in the abstract and the first three topics. for example:
in the abstract - line 14 "Primary tumors are rare, of which...". Line 15: "Patients with benign PCTs are usually asymptomatic and are found incidentally..." (the subject is the tumor, not the patient). Line 18: "varies" should spell "various". In paragraph 1 - line 35 "According to the 2015 WHO presented a new classification...", the first two words are unnecessary. etc.
Response 1: Thank you for your nice comment. We have carefully read the English expression in the article again, and have corrected the grammatical errors.
Point 2: In the description of the different imaging modalities, in the discussion about TEE, there is a need to better describe the added benefit it has over TTE. Fir which valves does it give a better picture, and why? I also suggest mentioning the better characterization of valvular stenosis and leaks, which is meaningful for deciding the necessity and timing of surgery.
While discussing MRI, I suggest the authors add a short explanation about T1, T2, SSFP sequencing, EGE and LGE, and preferably another sentence about special sequences such as fat-sat and tagging which are important for tumor identification.
Response 2: We thank the reviewer for this valuable comment. According to your suggestion, we have added the description about the superiority of TEE over TTE and a short explanation about T1, T2, SSFP sequencing, EGE and LGE.
Point 3: The paragraphs discussing each tumor are well written and easy to follow. The figures are informative and compliment the text. There is a brief description of major echo, CT and MRI findings. Cardiac PET is not discussed in all of the tumor types. I think that a table can be beneficial in summarizing the text (tumor type, common age and gender, common locations and appearance on echo, CT, MRI).
Response 3: Thank you very much for your comments. We agree that a table that summarized multimodality imaging features of benign PCTs can help readers better understand those tumors. And we have added the Table 1 to the revised manuscript.
Point 4: Although malignant primary cardiac tumors are extremely rare, and are not within the scope of this paper, I believe the authors should mention the common types, and reviewpapers discussing the characteristics of malignant primary cardiac tumors.
Response 4: We appreciate the constructive suggestion. Since our article mainly describes the multimodality imaging features of benign PCTs, we don’t mention malignant primary cardiac tumors. Malignant primary cardiac tumors include a variety of tumors, and we would like to discuss the characteristics of malignant primary cardiac tumors in our next article.

Round 2
Reviewer 1 Report
The authors did not reply satisfactorily to my criticisms.
The statements in point 2 and 5 are simply not true -according to my clinical experience- and the authors quote papers which do not tell what they are pretending. A single case report cannot make the history!
The English is still not satisfactory.